# Theoretical Studies on the Reaction Mechanism and Kinetics of Ethylbenzene-OH Adduct with O$_2$ and NO$_2$

Tingting Lu [1], Mingqiang Huang [1,*], Xin Lin [1], Wei Zhang [1], Weixiong Zhao [2], Changjin Hu [2], Xuejun Gu [2] and Weijun Zhang [2,*]

1   Fujian Province Key Laboratory of Modern Analytical Science and Separation Technology, College of Chemistry & Chemical Engineering and Environment, Minnan Normal University, Zhangzhou 363000, China; lutingt215@163.com (T.L.); 13850465812@163.com (X.L.); zhangwei1055088073@163.com (W.Z.)

2   Laboratory of Atmospheric Physico-Chemistry, Anhui Institute of Optics and Fine Mechanics, Chinese Academy of Sciences, Hefei 230031, China; wxzhaoaiofm@gmail.com (W.Z.); hucj@aiofm.ac.cn (C.H.); xjgu@aiofm.ac.cn (X.G.)

*   Correspondence: huangmingqiang@mnnu.edu.cn (M.H.); wjzhang@aiofm.ac.cn (W.Z.); Tel.: +86-596-2591-445 (M.H.); Fax: +86-596-2026-037 (M.H.)

**Abstract:** The OH-initiated reaction of ethylbenzene results in major OH addition, and the formed ethylbenzene-OH adducts subsequently react with O$_2$ and NO$_2$, which determine the components of the oxidation products. In this study, nine possible reaction paths of the most stable ethylbenzene-OH adduct, EB-Ortho (2-ethyl-hydroxycyclohexadienyl radical intermediate), with O$_2$ and NO$_2$ were studied using density functional theory and conventional transition state theory. The calculated results showed that ethyl-phenol formed via hydrogen abstraction was the major product of the EB-Ortho reaction with O$_2$ under atmospheric conditions. Peroxy radicals generated from O$_2$ added to EB-Ortho could subsequently react with NO and O$_2$ to produce 5-ethyl-6-oxo-2,4-hexadienal, furan, and ethyl-glyoxal, respectively. However, nitro-ethylbenzene formed from NO$_2$ addition to EB-Ortho was the predominant product of the EB-Ortho reaction with NO$_2$ at room temperature. The total calculated rate constant of the EB-Ortho reaction with O$_2$ and NO$_2$ was $9.57 \times 10^{-16}$ and $1.78 \times 10^{-11}$ cm$^3$ molecule$^{-1}$ s$^{-1}$, respectively, approximately equivalent to the experimental rate constants of toluene-OH adduct reactions with O$_2$ and NO$_2$. This study might provide a useful theoretical basis for interpreting the oxygen-containing and nitrogen-containing organics in anthropogenic secondary organic aerosol particles.

**Keywords:** ethylbenzene-OH adduct; reaction mechanism; rate constant; transition state theory

## 1. Introduction

Aromatic compounds emitted from anthropogenic sources, such as motor vehicle exhaust and solvent evaporation, are common volatile organic compounds in the urban atmosphere [1–3]. Aromatics discharged into the atmosphere mainly chemically react with OH radicals and nitrogen oxides to generate semi- and non-volatile oxygen-containing and nitrogen-containing organic products, which lead to the formation of secondary organic aerosol (SOA) via self-condensation and the gas/particle partitioning process [4–6]. The formed SOA particles are mainly fine particles (PM$_{2.5}$) with a particle size of less than 2.5 μm, which are the main components of urban photochemical smog. They scatter and absorb solar radiation, resulting in a reduced visibility of the atmosphere, and are a culprit in the formation of haze weather. This has become an important driving factor affecting radiative forcing and disturbing regional climate change [7,8]. There are many kinds of photooxidation products of aromatics, among which some oxygen-containing and nitrogen-containing organic products have a strong toxicity, such as epoxy carbonyl compounds, which could attack human gene tissue and have a strong carcinogenicity [9], as well as nitroaromatic compounds (NAC), which have mutagenicity and carcinogenicity [10,11],

and seriously endanger human health and have thus attracted extensive attention. Previous smog chamber experiments identified phenolic compounds, aldehydes, carboxylic acids, epoxy carbonyls, NACs, and other products [12–15]. However, the photooxidation reaction mechanisms of aromatics are complicated, and the study of the formation mechanism of epoxy carbonyls and NACs is imperative.

Ethylbenzene and other aromatic compounds are expected to be vital precursors of SOA in the presence of NO$x$ in an urban atmosphere [15–17]. Theoretical [18,19] and experimental [20,21] studies have indicated that 2-ethyl-hydroxylcyclohexadienyl radical (EB-Ortho) is the most stable ethylbenzene-OH adduct formed from OH addition to the ortho-position of the benzene ring. The resulting EB-Ortho then undergoes a hydrogen atom abstraction and addition reaction with O$_2$ and NO$x$ under atmospheric conditions. Atkinson and Arey [22] proposed the Path A mechanism, as shown in Figure 1. The O$_2$ molecule extracted a hydrogen atom gem to OH in EB-ortho to form HO$_2$ radical and 2-ethyl-phenol, which better explains the formation of phenolic products. Shepson et al. [23] and Klotz et al. [24] considered that EB-Ortho might react with O$_2$ molecules, as displayed in Path B of Figure 1, and the O$_2$ molecule extracted hydrogen of the OH group in EB-Ortho produced isomeric couple ethylbenzene oxide/oxepin (EB-Oxide and EB-oxepin). EB-oxide continued to react with OH radicals, O$_2$, and NO to yield epoxy carbonyls. This provided the theoretical basis for the formation of epoxy carbonyls. Suh et al. [25] then suggested Path C, as illustrated in Figure 1, where the O$_2$ molecule was added to the benzene ring to form the peroxy radical, which reacted with NO to generate the alkoxy radical. The alkoxy radical formed 5-ethyl-6-oxo-2,4-hexadienal (EB-iv) via aromatic ring cleavage and hydrogen extraction by O$_2$. In addition, the isomerization of the alkoxy radical formed chair-shaped intermediates, which reacted with O$_2$ and NO to produce furan and ethyl-glyoxal. These better explained the formation of aldehyde products formed from the photooxidation of aromatics.

**Figure 1.** Three suggested pathways of EB-Ortho with O$_2$ [22–25].

When NO$_2$ was present in the reaction system, Andino et al. [18] considered that the oxygen and nitrogen atom in the NO$_2$ molecule had a strong electronegativity, and a hydrogen extraction reaction might occur. The oxygen (or nitrogen) atom in the NO$_2$ molecule extracted hydrogen atom gem to OH in EB-Ortho, and Path D (or Path E), shown in Figure 2, occurred to produce HONO (or nitrous acid) and 2-ethyl-phenol. Similarly, the oxygen (or nitrogen) atom in the NO$_2$ molecule extracted the hydrogen of the OH

group in EB-Ortho, as illustrated in Path F (or Path G) of Figure 2, to yield HONO (or nitrous acid) and isomeric couple ethylbenzene oxide/oxepin. In addition, the oxygen atom in the $NO_2$ molecule added to the benzene ring, and Path H, displayed in Figure 2, occurred to form 2-hydroxyl-3-nitro-ethylbenzene after dehydrogenation. In addition, as illustrated in Path I of Figure 2, the nitrogen atom in the $NO_2$ molecule was added to the benzene ring to generate 2-nitro-ethylbenzene after dehydration. This provided the theoretical basis for the formation of phenolic products and NACs. However, the above theoretical studies [18,23–25] only focused on judging the feasibility of the reaction through the activation energy and the reaction energy, and rarely involved the calculation of the rate constant. Therefore, it is necessary to carry out an accurate calculation of the rate constant of each reaction channel, determine the main reaction path of the ethylbenzene-OH adduct with $O_2$ and $NO_2$, and better interpret the component information detected by the chamber experiments.

**Figure 2.** Six postulated channels of EB-Ortho with $NO_2$ [18].

Knispel et al. [26] considered the kinetics of the toluene-OH adduct reaction with $O_2$ and $NO_2$, and reported the corresponding rate constant of the ~5 × $10^{-16}$ and ~3 × $10^{-11}$ $cm^3$ molecule$^{-1}$ s$^{-1}$, respectively, although no experimental study has been performed on the kinetic experiments of ethylbenzene-OH adduct at present. Sato et al. [20] and Huang et al. [21] observed 5-ethyl-6-oxo-2,4-hexadienal, ethyl-glyoxal, furane, and 2-nitro-ethylbenzene with an aerodyne aerosol mass spectrometer, aerosol laser time-of-flight mass spectrometer, and vacuum ultraviolet photoionization mass spectrometer in ethylbenzene chamber experiments, respectively. However, no theoretical investigations on the reaction mechanism and rate constant of the ethylbenzene-OH adduct with $O_2$ and $NO_2$ molecules have been performed up until now. Our group carried out theoretical studies on the toluene-OH adduct with $O_2$ [27], OH radicals reaction with ethylbenzene [19], and m-xylene [28] using density functional theory (DFT) and conventional transition state theory (CTST). So, these methods were used to theoretically study the nine pathways of EB-Ortho with $O_2$ and $NO_2$ molecules in this work. The reaction energies and kinetic parameters in 298–398 K of different reaction paths were obtained to estimate the feasible reaction paths to propagate the consecutive reactions of the ethylbenzene-OH adduct.

## 2. Computational Methods

As suggested by Uc et al. [29] and shown in our previous studies [19,28], the methodology of BHandHLYP/6-311++G (d,p) followed by CCSD(T) calculation yields excellent results for H-aromatic reactions. The calculated rate coefficients obtained by CTST were approximately equivalent to the experimental values. Thus, the theoretical computational methods used in this study were similar to the protocol of our published studies [19,28].

All of the geometries and frequencies of the various species were carried out with a Gaussian 09 software package [30] at the BHandHLYP/6-311++G (d,p) level. The IRC [31] calculations were employed at the same level and basis set to verify the obtained transition states. High precision energies were operated on the CCSD(T)/6-311++G (d,p) level [32]. The rate coefficients of nine pathways were calculated using the CTST [33–35] with program of TheRate 1.0, which was provided by Zhang and Truong [36] on the csed.net website. According to the CTST under the condition of gas reaction [35], the rate constant, $k(T)$, was expressed by the following:

$$k(T) = \kappa \frac{k_B T}{h} \frac{Q_{TS}}{Q_R} e^{-Ea/K_B T}$$

where $\kappa$ is tunneling correction coefficient, $Q$ is the partition function of the transition states and the reactants, $k_B$ is Boltzmann's constant, and $Ea$ is the net activation energies. As reviewed by Vereecken et al. [37], the Wigner theory is the simplest methodology to describe tunneling, and it assumes an inverse parabolic barrier of a width determined by the imaginary wavenumber for the reaction coordinate saddle point. However, the Wigner theory does not yield a reliable tunneling correction, even when fitting the parabola to the actual potential energy surface profile [38]. The Eckart energy curve is obtained through the energy difference between the reactant, transition state, and product, and the width is implied by the imaginary frequency for movement along the reaction coordinates [39]. Eckart tunneling corrections have been found to yield results in good agreement with more elaborate treatments [40]. Thus, Eckart tunneling correction was selected in this study, which was computed using the Eckart method implemented in TheRate 1.1 program [36].

## 3. Result and Discussion

### 3.1. The Mechanism of EB-Ortho Adduct with $O_2$

According to our previous published study, $O_2$ added to EB-Ortho formed three possible isomers, and $O_2$ added to $C_3$ of a benzene ring producing the most stable ethyl-benzene peroxy radical, EB-PO3 [41]. In addition to the subsequent reactions of EB-PO3 to generate the bicyclic radical and epoxide radical, EB-PO3 could also react with nitrogen oxides to form ethylbenzene alkoxy radical EB-ii under atmospheric conditions [42]. So, the H atom abstraction and the possible subsequent reaction of EB-ii, as shown in Figure 1, were studied in first step. The obtained optimized geometries of various species and the potential energy surface of EB-Ortho reaction with $O_2$ are depicted in Figures 3–5 (Figure S1 in the Supplementary Information) and Figure 6, respectively.

(1) Path A: The H atom gem to OH in EB-Ortho could be abstracted by an electronegative atom of oxygen, and the corresponding transition structure of TS_A is displayed in Figure 3. TS_A was different from that of the $C_2H_5 + O_2$ abstraction reaction obtained by Ignatyev et al. [43], but similar to the one for the toluene-OH adduct with $O_2$ abstraction previously obtained in our previous study [27]. The C2•••H15 distance of TS_A was 1.170 Å smaller than that of H15•••O11 with a distance of 1.642 Å, indicating that abstraction occurred when the O atom was still somewhat far apart from the H atom. Compared with $O_2$, the O11•••O12 distance of TS_A was elongated to 0.045Å. The angle of O12•••O11•••H15 was 97.92°, the H atom nearly been shifted to the O atom, and the trend of the production of the HOO radical and 2-ethyl-phenol was presented. The transition vector (223 icm$^{-1}$) for hydrogen abstraction distinctly revealed the approach of the O atom to the H atom gem to OH in EB-Ortho. The results of the IRC showed that TS_A is an early transition structure, agreement with the remarkable exothermicity (29.08 kcal mol$^{-1}$, see Figure 6). The activation energy ($Ea$) for TS_A was 2.74 kcal mol$^{-1}$, a little less than that of the toluene-OH adduct with $O_2$ abstraction [27].

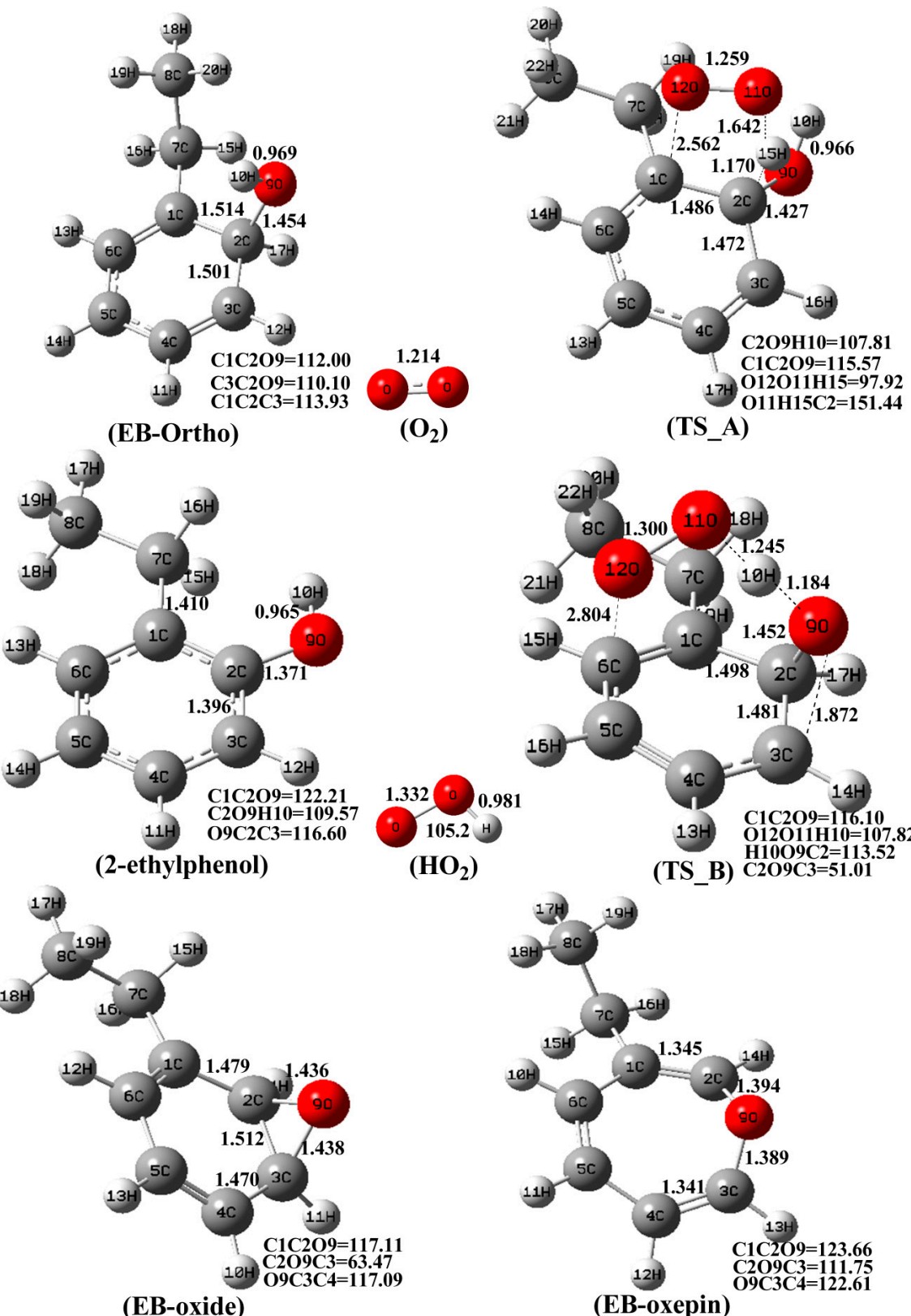

**Figure 3.** Optimized geometries of various species of Path A and Path B, as shown in Figure 1, at the BHandHLYP/6-311++G (d,p) level (Bond length: Å, Bond angle: °).

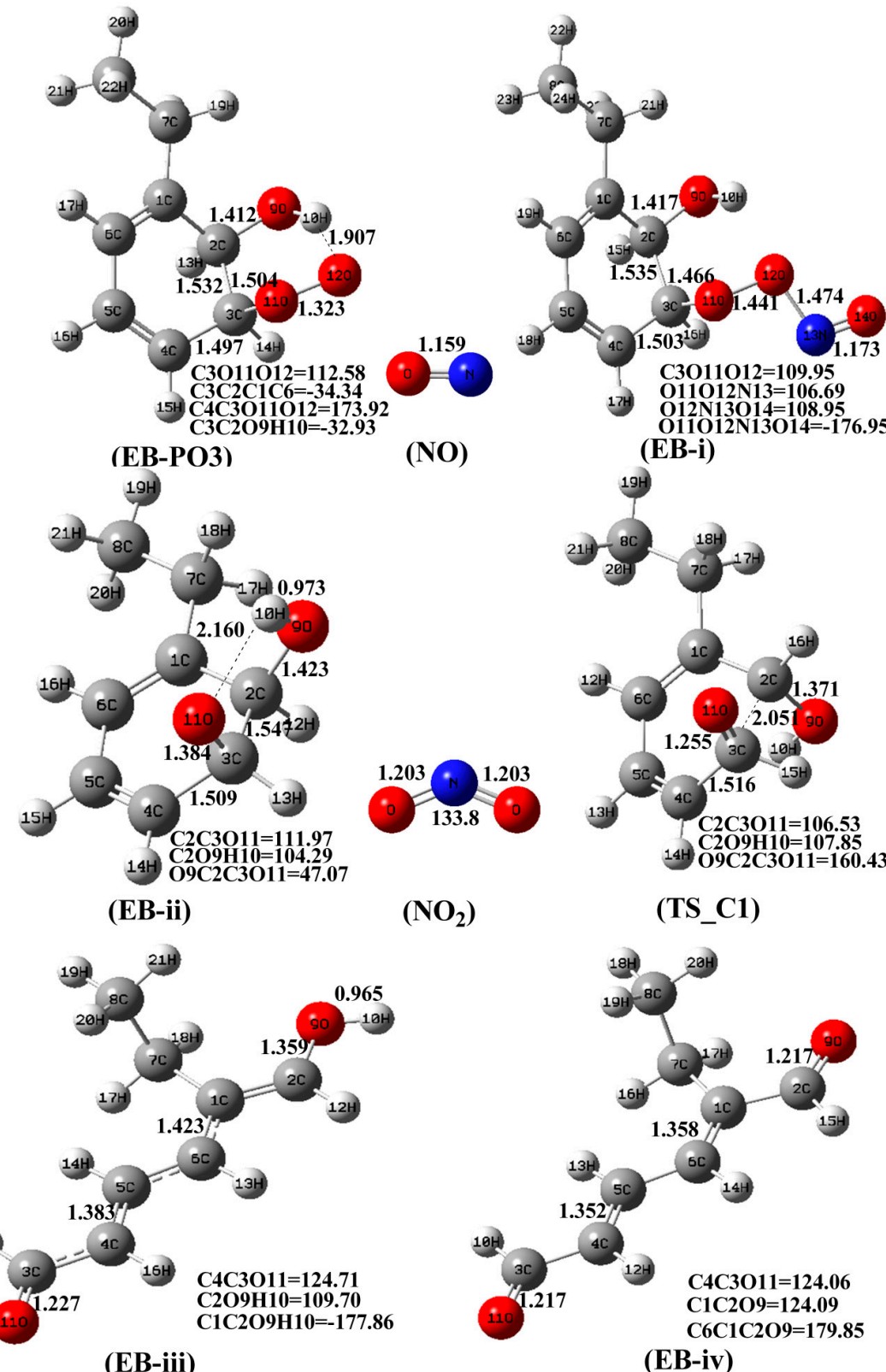

**Figure 4.** Optimized geometries of various species of Path C for the formation of 5-ethyl-6-oxo-2,4-hexadienal, as shown in Figure 1, at the BHandHLYP/6-311++G (d,p) level (Bond length: Å, Bond angle: °).

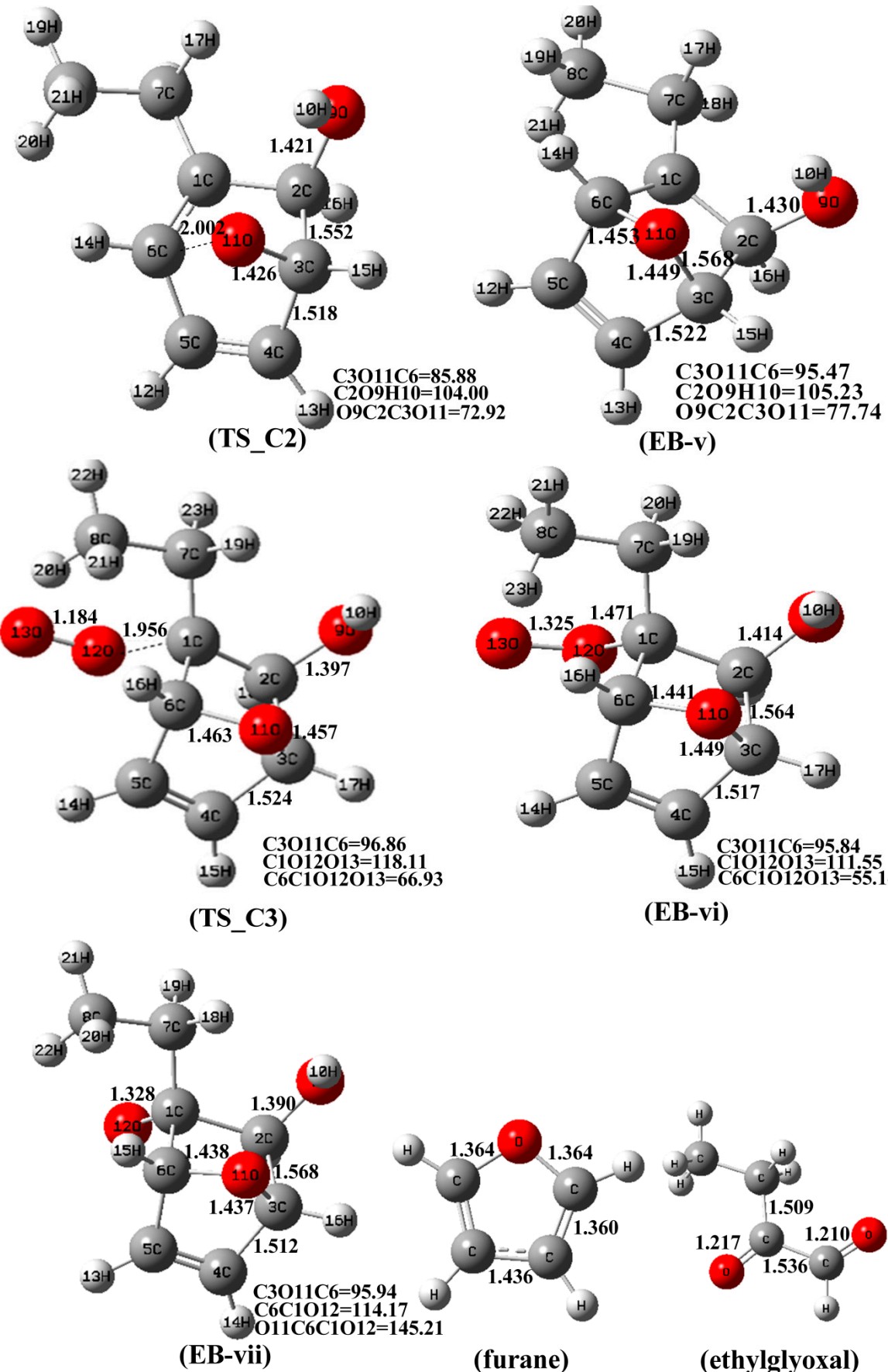

**Figure 5.** Optimized geometries of various species of Path C for the formation of furane and ethyl-glyoxal, as shown in Figure 1, at the BHandHLYP/6-311++G (d,p) level (Bond length: Å, Bond angle: °).

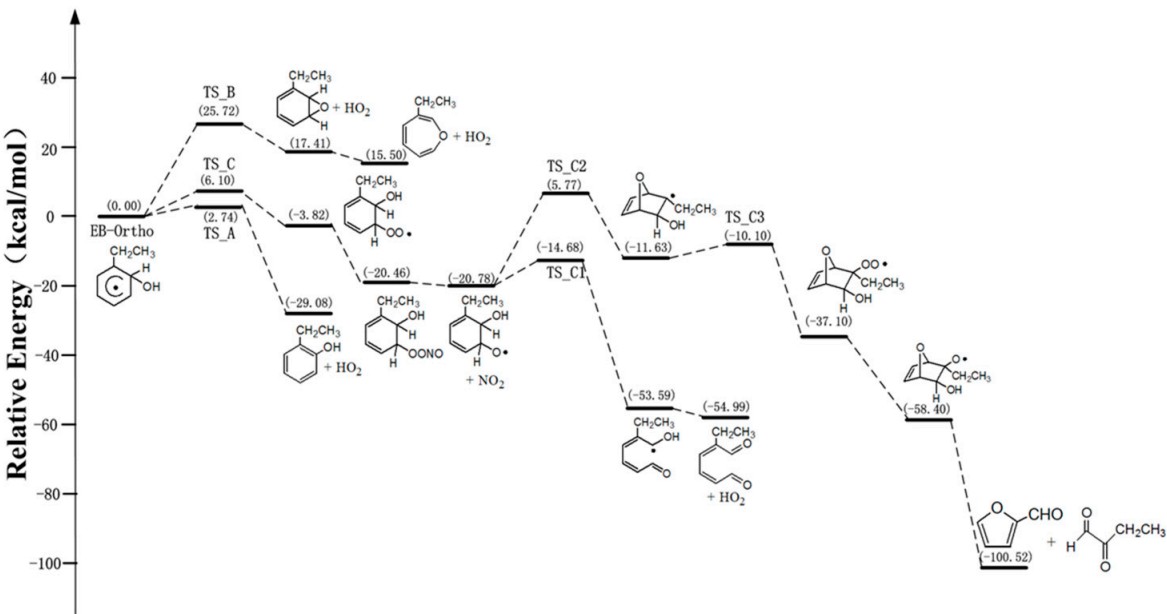

**Figure 6.** Potential energy surface at the CCSD(T)/6-311++G (d,p)//BHandHLYP/6-311++G (d,p) level of the EB-Ortho reaction with $O_2$, as shown in Figure 1.

(2) Path B: H atom abstraction from the OH group was proposed by Shepson and co-workers [23]. Klotz et al. [24] explored the abstraction reaction of the toluene-OH adduct with $O_2$, respectively. They proposed that the conceivable mechanism of this abstraction was the concerted process of hydrogen migration and the fragment of $HO_2$. Similarly, isomeric couple ethylbenzene oxide/oxepin (EB-Oxide and EB-oxepin) could probably form via H atom abstraction from the OH of EB-Ortho performed by $O_2$, and its corresponding transition state labeled as TS_B is displayed in Figure 3. For the process of EB-Ortho + $O_2$ → TS_B → EB-Oxide + HOO, the distance of O9•••H10 was stretched to 1.184Å of TS_B from 0.969 Å of EB-Ortho, and the distance of O11•••H10 was reduced to 0.981 Å of $HO_2$ from 1.245 Å of TS_B, indicating that the migration of the H atom from the OH to $O_2$ and dissociation of $HO_2$. The obtained imaginary frequency of TS_B was 1335 cm$^{-1}$, suggesting that TS_B was tighter than TS_A. The IRC calculation showed that TS_B was a late transition state, and the energy barrier of the reaction was as high as 25.72 kcal mol$^{-1}$, corresponding to the remarkable endothermic reaction (17.41 kcal mol$^{-1}$, see Figure 6).

(3) Path C: In a polluted urban atmosphere, the concentration of nitrogen oxides is significant [44]. Wallington et al. [45] showed that the ROO + NO reaction was generally fast. The formed peroxy radical EB-PO3 reacted with NO to generate the peroxynitrite adduct EB-i, with a reaction energy of −16.64 kcal mol$^{-1}$ (relative to the EB-PO3 and NO, see Figure 6). Subsequently, the dissociation of EB-i to the ethylbenzene alkoxy radical EB-ii and $NO_2$ needed only about 0.3 kcal mol$^{-1}$. Similar to peroxy radicals [41], the O atom of the alkoxy radical EB-iito formed a hydrogen bond with the H atom of OH, with a O11•••H10-O9 bond length of 2.160 Å (see Figure 4). From the alkoxy radical EB-ii, the β-fragmentation arose, with an activation energy of 6.10 kcal mol$^{-1}$, and the corresponding transition complex TS_C1 is displayed in Figure 4. The distance of C2...C3 was extended to 2.051 Å of TS_C1 from 1.547Å of EB-ii, indicating that C-C bond cleavage had occurred, and EB-iii produced with exoergic by 32.81 kcal mol$^{-1}$. From EB-iii, the second C=O of EB-iii could be generated by the H atom of the OH group abstraction performed by $O_2$. This step produced 5-ethyl-6-oxo-2,4-hexadienal (EB-iv), with a reaction energy of 1.40 kcal mol$^{-1}$ (see Figure 6).

As proposed by Jang and Kamens [13], the ring closure of EB-ii led to the formation of chair-shaped intermediates. The oxygen atom of the alkoxy radical EB-ii (O11) could attack on the benzene ring of carbon (C6), forming EB-v, and the corresponding transition state TS_C2, with a distance of C6●●●O11, was 2.002Å is displayed in Figure 5. The addition of $O_2$ to EB-v produced EB-vi via the relevant transition structure of TS_C3 with an energy barrier of 1.53 kcal mol$^{-1}$ (see Figure 6). EB-vi differed from EB-PO3 in that $O_2$ added to the different side of the aromatic ring as the OH group. Similar to EB-PO3, EB-vi reacted with NO to generate EB-vii and $NO_2$. The formed EB-vii then cleaved to form furan and ethyl-glyoxal, which were measured using previous ethylbenzene smog chamber experiments [20,21] with a significant exothermicity (42.12 kcal mol$^{-1}$, see Figure 6).

### 3.2. The Mechanism of EB-Ortho Adduct with $NO_2$

$NO_2$ derived fossil fuel combustion, motor vehicle exhaust emission, and other anthropogenic activity are the main pollutants in the atmosphere [46]. Field measurement results have shown that the content of $NO_2$ in the urban atmosphere is higher than that in suburban areas [47]. The daily mean concentration of $NO_2$ in some Chinese urban atmosphere can reach 150 μg m$^{-3}$ [48]. $NO_2$ could participate in atmospheric reactions of aromatic-OH adducts, thereby changing the components of anthropogenic SOA particles [49]. Under polluted atmospheric conditions, the concentration of $NO_2$ is relatively high [44], and the ethylbenzene-OH adduct reaction with $NO_2$ can not be ignored. Ethylbenzene smog chamber experiments have been used to measure the NACs products [20,21,50], but the relevant theoretical calculations have not been presently published. Andino et al. [18] suggested the reaction pathways of OH-aromatic adduct and $NO_2$, which generated a phenolic compound, aromatic oxide/oxepin via hydrogen abstraction (reaction D-G in Figure 2), and nitroaromatic compound (reaction H-I in Figure 2) through the addition reaction. To inspect the stabilities of the six reaction pathways of EB-Ortho and $NO_2$, optimization and frequency calculation of the various species were carried on BHandHLYP/6-311++G (d,p), and the potential energy surface and optimized geometries of various species are displayed in Figures 7–9 (Figure S2 in the Supplementary Information), respectively.

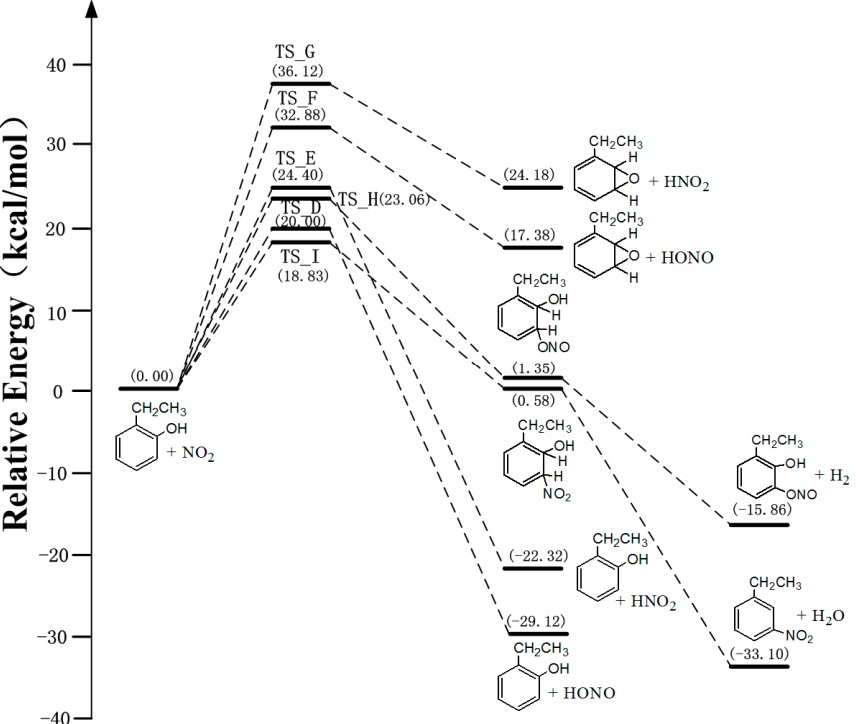

**Figure 7.** Potential energy surface at the CCSD(T)/6-311++G (d,p)//BHandHLYP/6-311++G (d,p) level of the EB-Ortho reaction with $NO_2$, as shown in Figure 2.

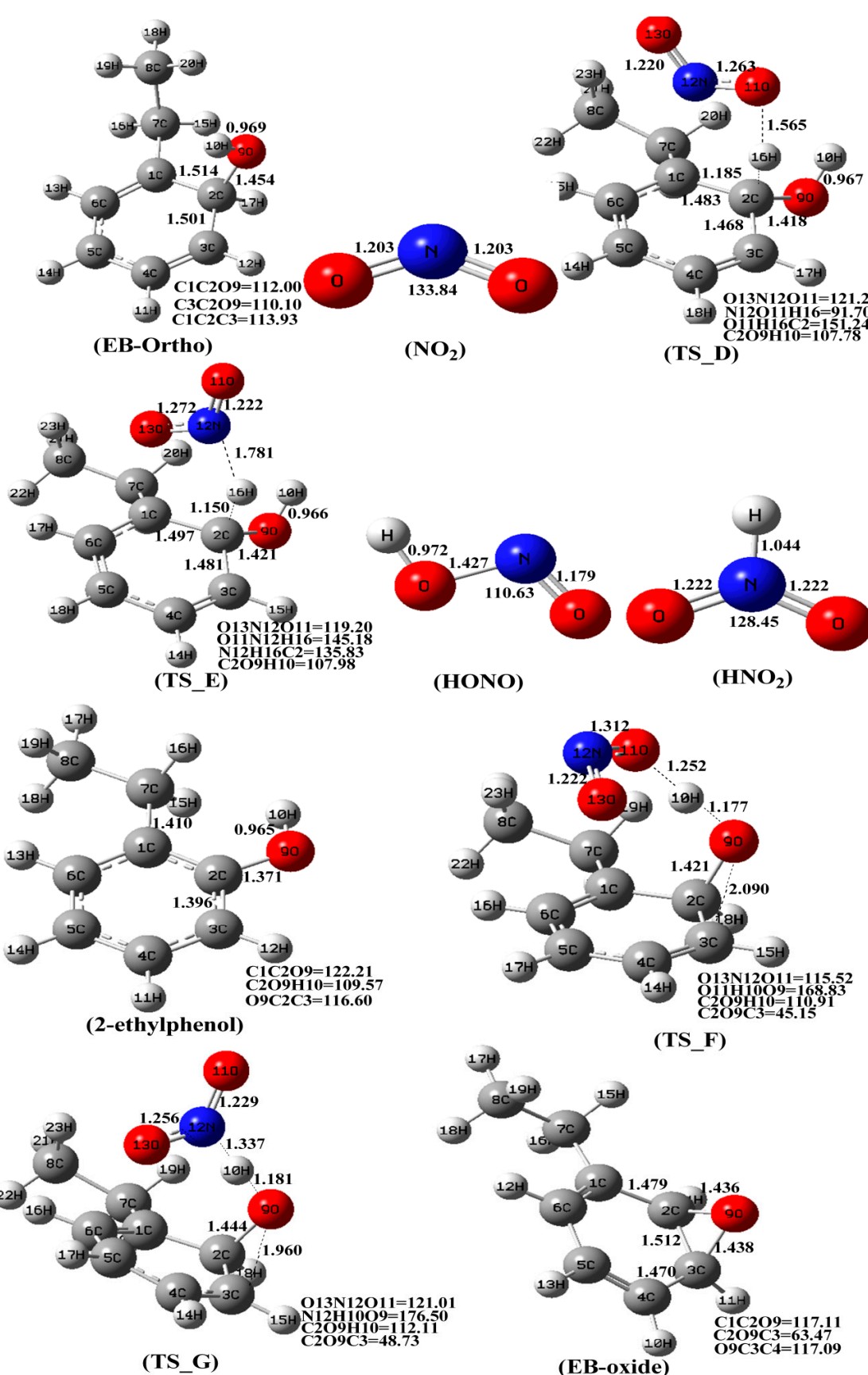

**Figure 8.** Optimized geometries of various species of Path D to Path G, as shown in Figure 2, the at BHandHLYP/6-311++G (d,p) level (Bond length: Å, Bond angle: °).

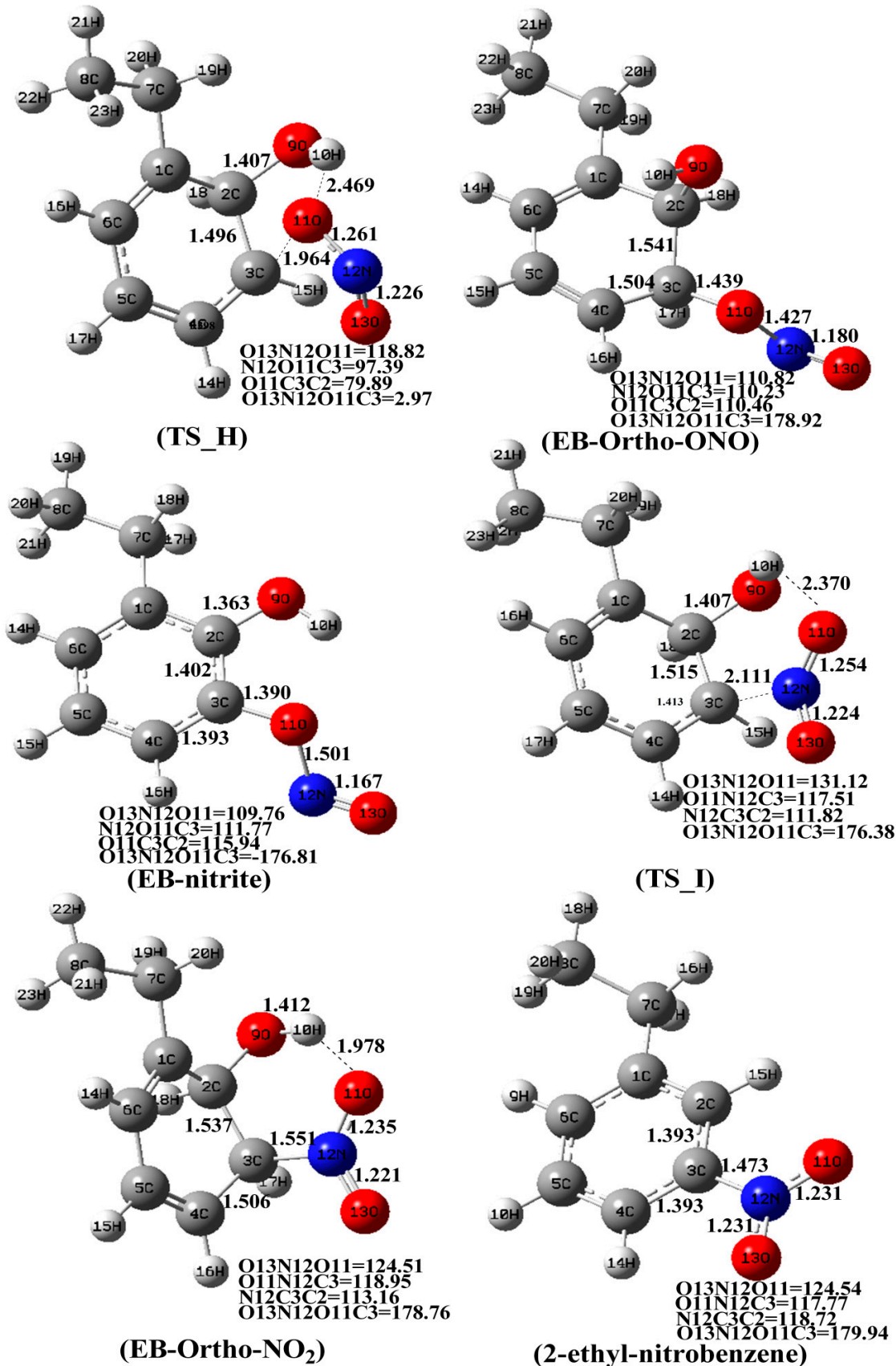

**Figure 9.** Optimized geometries of various species of Path H and Path I, as shown in Figure 2, at the BHandHLYP/6-311++G (d,p) level (Bond length: Å, Bond angle: °).

(1) Paths D−G: Both the nitrogen and oxygen atoms of $NO_2$ molecule contain lone pairs of electrons, which could abstract the hydrogen atom gem to or from the OH group in EB-Ortho. For the hydrogen abstraction, similar to $O_2$, $NO_2$ was also mainly extracted the hydrogen atom gem to the OH group in EB-Ortho, leading to the formation of 2-ethylphenol and HONO (or $HNO_2$), and the corresponding transition states TS_D and TS_E are depicted in Figure 8. However, the energy barrier for Reactions D and E was about 20–25 kcal mol$^{-1}$, greater than for Reaction A of 2.74 kcal mol$^{-1}$ (see Figures 6 and 7). These two reactions were exoergic by about 26 kcal mol$^{-1}$ (see Figure 7). On the contrary, the hydrogen atom from the OH group in the EB-Ortho abstraction needed to cross the high energy barrier (greater than 32 kcal mol$^{-1}$), with the relevant transition structures TS_F and TS_G displayed in Figure 8, corresponding to the significant endothermic reaction (greater than 17 kcal mol$^{-1}$, see Figure 7).

(2) Paths H−I: As proposed by Andino et al. [18], and Atkinson and Lloyd [51], $NO_2$ addition to EB-Ortho would be at the ortho position to the OH group. Both N and O atoms of the $NO_2$ molecule could attack the carbon (C3) of the benzene ring, forming EB-Ortho-ONO and EB-Ortho-$NO_2$, respectively. The corresponding transition states, TS_H and TS_I, with a distance of C3●●●O11 was 1.964 Å and C3●●●N12 was 2.111 Å, are illustrated in Figure 9, and the energy barrier of Reactions H and I was about 20 kcal mol$^{-1}$ (see Figure 7). It is worth noting that the O atom in the $NO_2$ and H atom of OH might have shaped the intramolecular hydrogen bond with a O11●●●H9-O10 bond length of 1.978 Å in EB-Ortho-$NO_2$, while EB-Ortho-ONO had no internal hydrogen bond (see Figure 9). In order to confirm the existence of the intramolecular hydrogen bond in EB-Ortho-$NO_2$, "Atoms In Molecules (AIM)" topological analysis [52] was performed at a BHandHLYP/6-311++G (d,p) level. According to the theory of AIM, the electron density ($\rho$) and Laplacian ($\nabla^2\rho$) were used to describe the strength and the characteristics of a bond, respectively. Here, $\nabla^2\rho = \lambda_1 + \lambda_2 + \lambda_3$, $\lambda_i$ was an eigenvalue of the Hessian matrix of $\rho$. When one of the three $\lambda_i$ was positive and the other two were negative, denoted as $(3, -1)$, it was called the bond critical point (BCP). When $\nabla^2\rho > 0$, the bond belonged to the covalent interactions, such as hydrogen bond, and van der Waals interactions [52]. As suggested by Lipkowski et al. [53], for the hydrogen bond of X-H●●●Y, there was a BCP between H●●●Y, and its electron density and Laplacian should be in the range of 0.002–0.035 and 0.02–0.139 a.u., respectively. A six-membered ring of O11●●●H10-O9-C2-C3-N12 was found in EB-Ortho-$NO_2$, there was a BCP between O11●●●H10, and the obtained electron density and Laplacian of the O11●●●H10-O9 bond in EB-Ortho-$NO_2$ was 0.026 and 0.082 a.u., respectively. These values fell within the proposed typical range of the hydrogen bond, suggesting that the O11●●●H10-O9 bond in EB-Ortho-$NO_2$ was an intramolecular hydrogen bond. However, no six-membered ring with O11●●●H10 was observed in EB-Ortho-ONO, and no BCP between O11●●●H10. Thus, no intramolecular hydrogen bond existed in EB-Ortho-ONO. It is clear that the intramolecular hydrogen bond played a role in stabilizing EB-Ortho-$NO_2$, which made the reaction energy of EB-Ortho-$NO_2$ (0.58 kcal mol$^{-1}$) lower than that of EB-Ortho-ONO (1.35 kcal mol$^{-1}$, see Figure 7). The reaction energy of EB-Ortho-$NO_2$ → 2-nitro-ethylbenzene + $H_2O$ was −33.10 kcal mol$^{-1}$ was greater than that of EB-Ortho-ONO → EB-nitrite + $H_2$ of −15.86 kcal mol$^{-1}$ (see Figure 7), indicating that the subsequent reaction of EB-Ortho-$NO_2$ was more likely to occur, and 2-nitro-ethylbenzene was the predominant product of EB-Ortho and $NO_2$. This was consistent with experimental evidence that existed for the nitroethylbenzen products of ethylbenzene smog chamber experiments [20,21,50].

### 3.3. Kinetics of Ethylbenzene-OH Adduct with $O_2$ and $NO_2$

The ambient temperature ranged from about 233 K to 313 K, owing to seasonal and regional variability [54]. However, the previous experiments mainly measured the rate constant of aromatic-OH adduct and $O_2$ or $NO_2$ in the range of 298–398 K [26]. In order for a better comparison with the experimental values, according to CTST, rate coefficients of the Reactions A−I were calculated with the program TheRate 1.0 [36] in the range of 298–398 K.

The obtained rate constants and Eckart coefficients are listed in Tables 1 and 2. Under atmospheric conditions (298 K in Table 1), the total rate coefficient of EB-Ortho and $O_2$ was $9.57 \times 10^{-16}$ cm$^3$ molecule$^{-1}$ s$^{-1}$, slightly higher than the experimental measurements of the toluene-OH adduct and $O_2$ reaction of the ~$5 \times 10^{-16}$ cm$^3$ molecule$^{-1}$ s$^{-1}$ [26]. The rate coefficient of Reactions A and C was about $10^{11}$–$10^{15}$ times greater than Reaction B in 298–398 K, and the rate coefficient of Reaction B was only $2.47 \times 10^{-31}$ cm$^3$ molecule$^{-1}$ s$^{-1}$ at room temperature, indicating that the formation of ethylbenzene oxide/oxepin via Reaction B almost did not occur in the atmosphere. Reaction paths A and C had similar energy barrier heights, but reaction A, with a significant exothermicity, was easy to carry out under atmospheric conditions. This result was agreement with the yield of ethyl-phenol that was detected in previously published ethylbenzene smog chamber experiments [20,21]. The rate constant of Reaction C was about 1/30 of Reaction A in the 298–398 K, the subsequent reactions of the formed peroxy radical EB-PO3 leading to the formation of 5-ethyl-6-oxo-2,4-hexadienal, furan, and ethyl-glyoxal. However, there were many reaction steps to form these products, and their yields were relatively low compared with the ethyl-phenol, which was confirmed by the results of experimental photooxidation of ethylbenzene. Sato et al. [20], and Huang and co-workers [21] detected only a small amount of 5-ethyl-6-oxo-2,4-hexadienal, ethyl-glyoxal, and furane with an aerodyne aerosol mass spectrometer, aerosol laser time-of-flight mass spectrometer, and vacuum ultraviolet photoionization mass spectrometer, respectively.

**Table 1.** Partial and overall rate constants (cm$^3$ molecule$^{-1}$ s$^{-1}$) and Eckart coefficients as a function of temperature of EB-Ortho reaction with $O_2$.

| T (K) | Reaction A | κ | Reaction B | κ | Reaction C | κ | Overall |
|---|---|---|---|---|---|---|---|
| 298 | $9.37 \times 10^{-16}$ | 1.15 | $2.47 \times 10^{-31}$ | 6.53 | $2.04 \times 10^{-17}$ | 1.19 | $9.57 \times 10^{-16}$ |
| 308 | $1.07 \times 10^{-15}$ | 1.14 | $8.44 \times 10^{-31}$ | 5.79 | $2.62 \times 10^{-17}$ | 1.18 | $1.10 \times 10^{-15}$ |
| 318 | $1.22 \times 10^{-15}$ | 1.14 | $2.69 \times 10^{-30}$ | 5.20 | $3.32 \times 10^{-17}$ | 1.17 | $1.25 \times 10^{-15}$ |
| 328 | $1.37 \times 10^{-15}$ | 1.13 | $8.03 \times 10^{-30}$ | 4.73 | $4.16 \times 10^{-17}$ | 1.17 | $1.41 \times 10^{-15}$ |
| 338 | $1.54 \times 10^{-15}$ | 1.13 | $2.26 \times 10^{-29}$ | 4.33 | $5.14 \times 10^{-17}$ | 1.16 | $1.59 \times 10^{-15}$ |
| 348 | $1.72 \times 10^{-15}$ | 1.12 | $6.02 \times 10^{-29}$ | 4.01 | $6.26 \times 10^{-17}$ | 1.15 | $1.78 \times 10^{-15}$ |
| 358 | $1.92 \times 10^{-15}$ | 1.12 | $1.53 \times 10^{-28}$ | 3.73 | $7.62 \times 10^{-17}$ | 1.15 | $2.00 \times 10^{-15}$ |
| 368 | $2.13 \times 10^{-15}$ | 1.11 | $3.69 \times 10^{-28}$ | 3.49 | $9.15 \times 10^{-17}$ | 1.14 | $2.22 \times 10^{-15}$ |
| 378 | $2.35 \times 10^{-15}$ | 1.11 | $8.55 \times 10^{-28}$ | 3.29 | $1.09 \times 10^{-16}$ | 1.14 | $2.46 \times 10^{-15}$ |
| 388 | $2.58 \times 10^{-15}$ | 1.11 | $1.90 \times 10^{-27}$ | 3.12 | $1.29 \times 10^{-16}$ | 1.13 | $2.71 \times 10^{-15}$ |
| 398 | $2.83 \times 10^{-15}$ | 1.10 | $4.08 \times 10^{-27}$ | 2.97 | $1.51 \times 10^{-16}$ | 1.13 | $2.98 \times 10^{-15}$ |

**Table 2.** Partial and overall rate constants (cm$^3$ molecule$^{-1}$ s$^{-1}$) and Eckart coefficients as a function of temperature of EB-Ortho reaction with $NO_2$.

| T (K) | Reaction D | κ | Reaction E | κ | Reaction F | κ | Reaction G | κ | Reaction H | κ | Reaction I | κ | Overall |
|---|---|---|---|---|---|---|---|---|---|---|---|---|---|
| 298 | $3.46 \times 10^{-16}$ | 0.072 | $9.37 \times 10^{-16}$ | 1.15 | $6.63 \times 10^{-18}$ | 2.61 | $4.77 \times 10^{-22}$ | 3.13 | $6.37 \times 10^{-12}$ | 0.13 | $1.14 \times 10^{-11}$ | 0.33 | $1.78 \times 10^{-11}$ |
| 308 | $3.55 \times 10^{-16}$ | 0.10 | $1.07 \times 10^{-15}$ | 1.14 | $8.52 \times 10^{-18}$ | 2.47 | $7.77 \times 10^{-22}$ | 2.92 | $6.54 \times 10^{-12}$ | 0.18 | $1.20 \times 10^{-11}$ | 0.42 | $1.85 \times 10^{-11}$ |
| 318 | $3.63 \times 10^{-16}$ | 0.15 | $1.22 \times 10^{-15}$ | 1.14 | $1.08 \times 10^{-17}$ | 2.36 | $1.24 \times 10^{-21}$ | 2.75 | $6.71 \times 10^{-12}$ | 0.26 | $1.26 \times 10^{-11}$ | 0.54 | $1.93 \times 10^{-11}$ |
| 328 | $3.72 \times 10^{-16}$ | 0.21 | $1.37 \times 10^{-15}$ | 1.13 | $1.36 \times 10^{-17}$ | 2.26 | $1.92 \times 10^{-21}$ | 2.60 | $6.88 \times 10^{-12}$ | 0.35 | $1.32 \times 10^{-11}$ | 0.68 | $2.01 \times 10^{-11}$ |
| 338 | $3.82 \times 10^{-16}$ | 0.29 | $1.54 \times 10^{-15}$ | 1.13 | $1.69 \times 10^{-17}$ | 2.17 | $2.92 \times 10^{-21}$ | 2.47 | $7.07 \times 10^{-12}$ | 0.47 | $1.39 \times 10^{-11}$ | 0.85 | $2.10 \times 10^{-11}$ |
| 348 | $3.92 \times 10^{-16}$ | 0.39 | $1.72 \times 10^{-15}$ | 1.12 | $2.09 \times 10^{-17}$ | 2.10 | $4.35 \times 10^{-21}$ | 2.36 | $7.26 \times 10^{-12}$ | 0.62 | $1.46 \times 10^{-11}$ | 1.04 | $2.19 \times 10^{-11}$ |
| 358 | $4.03 \times 10^{-16}$ | 0.51 | $1.92 \times 10^{-15}$ | 1.12 | $2.55 \times 10^{-17}$ | 2.03 | $6.36 \times 10^{-21}$ | 2.27 | $7.46 \times 10^{-12}$ | 0.80 | $1.53 \times 10^{-11}$ | 1.25 | $2.28 \times 10^{-11}$ |
| 368 | $4.13 \times 10^{-16}$ | 0.67 | $2.13 \times 10^{-15}$ | 1.11 | $3.09 \times 10^{-17}$ | 1.97 | $9.14 \times 10^{-21}$ | 2.18 | $7.67 \times 10^{-12}$ | 1.03 | $1.60 \times 10^{-11}$ | 1.50 | $2.37 \times 10^{-11}$ |
| 378 | $4.25 \times 10^{-16}$ | 0.86 | $2.35 \times 10^{-15}$ | 1.11 | $3.72 \times 10^{-17}$ | 1.92 | $1.29 \times 10^{-20}$ | 2.11 | $7.88 \times 10^{-12}$ | 1.31 | $1.67 \times 10^{-11}$ | 1.79 | $2.46 \times 10^{-11}$ |
| 388 | $4.37 \times 10^{-16}$ | 1.10 | $2.58 \times 10^{-15}$ | 1.11 | $4.44 \times 10^{-17}$ | 1.87 | $1.80 \times 10^{-20}$ | 2.05 | $8.10 \times 10^{-12}$ | 1.64 | $1.75 \times 10^{-11}$ | 2.10 | $2.56 \times 10^{-11}$ |
| 398 | $4.49 \times 10^{-16}$ | 1.38 | $2.83 \times 10^{-15}$ | 1.10 | $5.27 \times 10^{-17}$ | 1.83 | $2.47 \times 10^{-20}$ | 1.99 | $8.33 \times 10^{-12}$ | 1.88 | $1.84 \times 10^{-11}$ | 2.46 | $2.67 \times 10^{-11}$ |

The rate coefficient of Reaction I, which led to the formation of 2-nitro-ethylbenzene, was the largest among Reactions A−I in the 298–398 K (see Table 2). The total rate coefficient of EB-Ortho and $NO_2$ was $1.78 \times 10^{-11}$ cm$^3$ molecule$^{-1}$ s$^{-1}$ at room temperature, approximately equivalent to the experimental value of the OH-toluene reaction with $NO_2$ of ~$3 \times 10^{-11}$ cm$^3$ molecule$^{-1}$ s$^{-1}$ [26]. This coefficient was about $10^4$ times greater than that of Reaction A in 298–398 K. It is worth noting that the ROO + NO reaction was generally fast. At room temperature, the rate constant for the reaction between $CH_3OO$ and NO

was determined to be $(3.0 \pm 0.2) \times 10^{-12}$ cm$^3$ molecule$^{-1}$ s$^{-1}$ by Adachi and Basco [55] using flash photolysis and kinetic spectroscopy. As reviewed by Wallington et al. [45], the rate coefficient of ROO and NO was about $10^{-12}$ cm$^3$ molecule$^{-1}$ s$^{-1}$ at room temperature. However, the calculated rate constant of the reaction of EB-Ortho and O$_2$ to form the peroxy radical was $2.04 \times 10^{-17}$ cm$^3$ molecule$^{-1}$ s$^{-1}$. Thus, the total rate coefficient of the consecutive reactions of EB-Ortho, O$_2$, and NO was about $10^{-29}$ cm$^3$ molecule$^{-1}$ s$^{-1}$ at room temperature, which was far less than that of the total rate coefficient of EB-Ortho and NO$_2$ ($1.78 \times 10^{-11}$ cm$^3$ molecule$^{-1}$ s$^{-1}$). So, the ethylbenzene-OH adduct reaction with NO$_2$ is the most important reaction channel in the contaminated atmosphere with a high concentration of NO$x$. The measurements of nitro-ethylbenzene products in an ethylbenzene smog chamber [20,21,50] also confirmed these results. In addition, the total rate coefficient of O atom of the NO$_2$ molecule extracted hydrogen from the OH group in EB-Ortho (Reaction F) was $6.63 \times 10^{-18}$ cm$^3$ molecule$^{-1}$ s$^{-1}$ under atmospheric conditions, which was about 1/10 of Reaction C of EB-Ortho and O$_2$ in the 298–398 K. Thus, ethylbenzene oxide/oxepin might form via H atom abstraction from the OH of EB-Ortho performed by NO$_2$ in a polluted atmosphere. This could provide the theoretical basis for the epoxide products formed from the subsequent reactions of ethylbenzene oxide/epoxide observed in the ethylbenzene smog chamber experiments [12].

It is worth noting that the concentration of O$_2$ in the urban atmosphere was much higher than that of NO$_2$, and the EB-Ortho reaction with the O$_2$ of pathway A shown in Figure 1 was the main reaction channel under atmospheric conditions. However, as EB-Ortho and NO$_2$ had a faster reaction rate, the reaction path I displayed in Figure 2 became increasingly significant with the increasing concentration of NO$_2$. The experimental chamber results of Nishino et al. [56] demonstrated that aromatic-OH adducts reacted with O$_2$ and NO$_2$ were of equal importance when the concentration of NO$_2$ reached about 3.3 ppm for toluene. The ambient concentrations of toluene and its nitrated products measured in Los Angeles were consistent with their laboratory measurements. Nitroaromatic compounds would be the main component of anthropogenic SOA particles in the background of high concentration of NO$x$ in the urban atmosphere.

## 4. Conclusions

Here, O$_2$ and NO$_2$ abstraction and addition with ethylbenzene-OH adduct has been studied in 298–398 K using DFT and CTST methods. Ethyl-phenol formed via the hydrogen atom gem to OH abstracted by O$_2$ was the major product of the ethylbenzene-OH adduct reaction with O$_2$ under atmospheric conditions, while nitro-ethylbenzene formed through NO$_2$ addition was the predominant product of the ethylbenzene-OH adduct reaction with NO$_2$ at room temperature. Ethyl-phenol, 5-ethyl-6-oxo-2,4-hexadienal, furan, ethyl-glyoxal, and nitro-ethylbenzene, which were measured in ethylbenzene smog chamber experiments, were theoretically explained in detail in this study. The total rate coefficient of the O$_2$ and NO$_2$ reaction with ethylbenzene-OH was $9.57 \times 10^{-16}$ and $1.78 \times 10^{-11}$ cm$^3$ molecule$^{-1}$ s$^{-1}$, respectively, slightly higher than the experimental measurements of the toluene-OH reaction with O$_2$ and NO$_2$. These were helpful for understanding the formation mechanism of oxygen-containing and nitrogen-containing organics formed from the photooxidation of ethylbenzene in the presence of NO$x$ in an urban atmosphere.

**Supplementary Materials:** The following are available online at https://www.mdpi.com/article/10 .3390/atmos12091118/s1, Figure S1: Optimized geometries of various species of EB-Ortho react with O$_2$ at BHandHLYP/6-311++ G (d,p) level (Bond length: Å, Bond angle: (°)). Figure S2: Optimized geometries of various species of EB-Ortho react with NO$_2$ at BHandHLYP/6-311++ G (d,p) level (Bond length: Å, Bond angle: (°)).

**Author Contributions:** Conceptualization, M.H. and W.Z. (Weijun Zhang); methodology, W.Z. (Weijun Zhang) and X.G.; software, X.G. and C.H.; validation, M.H. and W.Z. (Weijun Zhang); formal analysis, W.Z. (Weixiong Zhao); data curation, T.L. and X.L.; writing—original draft prepara-



tion, T.L., X.L., and W.Z. (Wei Zhang); writing—review and editing, M.H.; project administration, M.H.; funding acquisition, M.H. All of the authors reviewed and commented on the paper. All authors have read and agreed to the published version of the manuscript.

**Funding:** This work is supported by the National Natural Science Foundation of China (no. 41575118), the Key Project of Natural Science Foundation of Fujian Province of China (no. 2020J02044) and the advanced science and technology project of Minnan Normal University (no. 4201-L11805). The authors thank Michael Nusbaum from the Department of English, Xiamen University, Tan Kah Kee College for the language edits. The authors also express our gratitude to the referees for their valuable comments.

**Institutional Review Board Statement:** Not applicable.

**Informed Consent Statement:** Not applicable.

**Data Availability Statement:** Not applicable.

**Acknowledgments:** Schemes in Figures 1 and 2 are consulted the corresponding references and completely redrawn and significantly changed beyond recognition. We thank the authors of these references for providing us with the corresponding schematic diagram of the reaction mechanism. Also, we thank Michael Nusbaum from Department of English, Xiamen University, Tan Kah Kee College for language edits, and express our gratitude to the referees for their valuable comments.

**Conflicts of Interest:** The authors declare no conflict of interest.

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
