# Peer review of "Theoretical Studies on the Reaction Mechanism and Kinetics of Ethylbenzene-OH Adduct with O2 and NO2"

_atmosphere, doi:10.3390/atmos12091118_

Round 1

Reviewer 1 Report

Ethylbenzene was expected to be vital precursor of secondary organic aerosol (SOA) in the presence of NOx in urban atmosphere. The present work investigates on the 9 reaction paths and rate constants of the most stable ethylbenzene-OH adduct EB-Ortho with O2 and NO2 by both high level quantum chemical calculations and the transition state theory, indicating that ethylphenol formed via the hydrogen abstraction is the major product of EB-Ortho reaction with O2, while, nitro-ethylbenzene form from NO2 addtion to EB-Ortho is the predominant product of EB-Ortho reaction with NO2 under atmospheric conditions. With the help of high level theoretical methods, the conclusions drawn by this study could contribute to deepen our understanding on the mechanism of oxygen-containing and nitrogen-containing organics in aromatic SOA particles in atmosphere. I would like to recommend the publication of this manuscript on the journal of Atmosphere after addressing flowing questions.

  1. Line 100, there was two different tunneling corrections for the rate constant, Wigner and Eckart. Why Eckart tunneling correction was seclected in this study? Please clarify.

  1. Line 194-196, why the O11... H9-O10 bond in the intermediate of EB-Ortho-NO2 was considered as intramolecular hydrogen bond. Please justify. Compared with the intermediate of EB-Ortho-ONO, what is the role of the intramolecular hydrogen bond?

  1. Line 210-212, why the temperature range of 298–398 K was chosen for rate coefficients of the Reaction A-I? Please describe.

  1. Line 240-245 the obtained total rate coefficient of EB-Ortho and NO2 is 1.78×10-11 cm3 molecule-1 s-1 at the room temperature, which was about 104 times greater than that the reaction of EB-Ortho and O2. However, ROO + NO reaction was generally fast (Line 146), I would doubt the ethylbenzene-OH adduct reaction with NO2 was the most important reaction channel in contaminated atmosphere. More discussions or comments on this should help to put off readers's doubts.
  2. The language of this manuscript was poor. I found some grammatical errors throughout the manuscript. Therefore, the quality of the English on the paper should be improved with the help of English native writer.

MINOR COMMENTS

L23 “subsequent complex react with” change to “subsequently complex react with”

L35 “addtion” change to “addition”

L48-49 “the study of the chemical composition and formation mechanism of aromatic SOA are imperative” – correct

L79-80 “Our group have carried out” – correct

L89-90 “equivalent with the experimental values” change to “equivalent to the experimental values”

Line 164 “NO2”, Line 183, “kcal mol-1”, and so on, please note the superscripts and subscripts of the whole manuscript.  

Line 233 “Reactio” change to “Reaction”

Line 241 “approximate equivalent with” – correct

Line 242 “OH-tolune” change to “OH-toluene”

Line 257 “addtion” change to “addition”

Line 263 “foramtion” change to “formation”

Line

Reviewer 2 Report

Lu et al. present a theoretical study of the reaction of ethylbenzene-OH adducts, a product of gas-phase oxidation initiated by the OH radical, with O2 and NO2 molecules. The authors use density functional theory (DFT) and classical transition state theory (CTST) to study the reaction pathways and kinetics. This is an interesting study relevant to atmospheric chemistry. The authors would benefit from a professional proofreading and English-language writing service. I do not recommend publication of the paper, and I offer my rationale for the recommendation below. The study is nevertheless both interesting and relevant. I hope that the authors will re-submit this study in some form after major revisions to the paper, and I would be happy to review the paper once more.

Major comments

[1] The introductory paragraph reflects poor background knowledge of atmospheric chemistry.

[2] The discussion of mechanisms centers on O2 and NO2 reactions, which are well-known and have been summarized in various reviews and textbooks. The authors should acknowledge this fact and then provide a strong case that their work is novel. The authors should identify the need for their specific contribution, quantitatively if possible.

[3] The writing is poor. I recommend improving the following:

* organization of sentences within each paragraph (especially in the results section)

* explanation of the figures in the captions and text

* linkage between the figures and text

[4] The figures are of poor quality

* the print quality is poor, e.g., the font is unreadable, the image has poor resolution

* there is a great deal of superfluous detail

[5] The translation to English is poor. These problems are encountered at least once in every sentence:

* grammatical error

* unconventional word choice

* invented term

* ambiguous phrase

Technical corrections

I provide a few technical corrections here. However, I did not proofread the entire paper. I recommend a professional proofreading service.

-- Title and author list –

Title: don’t capitalize “with”

Two instances of a missing space between “[email protected]” and “(M.H.);”

--  abstract --

Lines15 and 16 – this sentence is hard to understand. Do you mean “adducts subsequently react with O2 and NO2” ?

Line 18 “using density” …

Move “in this study” to the beginning of the sentence

Line 20 “the EB-Ortho reaction”

Line 22 remove “complex”

Line 22 remove space after diene-

Line 22 comma behind furan

Line 23 “formed”

Line 24 “at room “

26 “approximately equivalent to”

27 “a useful”

28 write out “secondary organic aerosol”

-- main text --

References 1 through 6 are cited in error. Reading the literature will reveal that these facts should be attributed to older papers.

The fact that aromatics are inherently toxic has been known for over half a century; I find it difficult to imagine that the discussion has not evolved since then.

Line 59-60 This sentence is hard to understand, specifically “may occur subsequent” – do you mean, “may occur, and then subsequently” or “may occur after” ?

Line 74 This sentence is incomplete, maybe you want to say “respectively, although no experimental study has been performed on the”

Line 77-78 since you do not use these acronyms or methods, I would remove the acronyms. Also, the acronyms ALTOFMS and VUV-PIMS are not correct.

Line 84 it is not clear what “respectively” means here

Line 86 Change “got” to obtained

Line 164 I think you mean “chair-shaped” and not “wheelchair shaped”

-- Figures --

In Figures 1 and 2, pathways A through I are outlined. Each of these pathways should be described in the text. This is typical for a mechanistic study. Further, adding labels to the pathways implies that you will refer to the labels in the discussion.

Figures 1 and 2 may appear too early in the text. Where the figures are discussed, a simplified schematic may be helpful.

Figures 1 and 2 do not appear to correspond to any particular study. Citations are necessary for credibility. If the mechanisms are entirely original, then they should appear later in the text with “results and discussion.”

Figure 3 is rendered in poor quality. Please provide a higher quality figure so that the text in the picture is readable.

The text in Figure 3 is too small.

Figure 3 contains more configurations than are discussed. It would be better to break Figure 3 into smaller figures to be referred to during the discussion. The entire figure is helpful as a supplement to the paper.

The caption of Figure 4 would be more helpful if it referred the reader to back to Figure 1. This would help the reader understand the meaning of the labels TS_A, TS_B, and TS_C.

Reviewer 3 Report

Paper looks interesting and important, however needs additional aspects interesting for readers (see review)

Round 2

Reviewer 2 Report

I commend the authors for an excellent job responding to comments, especially with regard to updating the background material and communicating the justification for the present study. I am pleased to recommend the publication of this study.